# LncRNA *PART1* Promotes Proliferation and Migration, Is Associated with Cancer Stem Cells, and Alters the miRNA Landscape in Triple-Negative Breast Cancer

**DOI:** 10.3390/cancers13112644

**Published:** 2021-05-27

**Authors:** Brianne M. Cruickshank, Marie-Claire D. Wasson, Justin M. Brown, Wasundara Fernando, Jaganathan Venkatesh, Olivia L. Walker, Fiorella Morales-Quintanilla, Margaret L. Dahn, Dejan Vidovic, Cheryl A. Dean, Carter VanIderstine, Graham Dellaire, Paola Marcato

**Affiliations:** 1Department of Pathology, Faculty of Medicine, Dalhousie University, Halifax, NS B3H 4R2, Canada; brianne.cruickshank@dal.ca (B.M.C.); mwasson@dal.ca (M.-C.D.W.); justin.brown@dal.ca (J.M.B.); wasufer@Dal.Ca (W.F.); jg664568@dal.ca (J.V.); ol458762@dal.ca (O.L.W.); meg.dahn@dal.ca (M.L.D.); dejan.vidovic@dal.ca (D.V.); deanc@dal.ca (C.A.D.); cr399192@dal.ca (C.V.); dellaire@dal.ca (G.D.); 2Department of Biology, Faculty of Science, Saint Mary’s University, Halifax, NS B3H 3C3, Canada; fiorella.morales@smu.ca; 3Department of Biochemistry and Molecular Biology, Faculty of Medicine, Dalhousie University, Halifax, NS B3H 4R2, Canada; 4Department of Microbiology and Immunology, Faculty of Medicine, Dalhousie University, Halifax, NS B3H 4R2, Canada

**Keywords:** lncRNA, miRNAs, TNBC, *PART1*, CSCs

## Abstract

**Simple Summary:**

Long non-coding RNAs (lncRNAs) play an important role in cancer progression. Herein we provide new information regarding the role of prostate androgen regulated transcript 1 (*PART1*). We show that the lncRNA *PART1* is enriched in triple-negative breast cancers and cancer stem cell populations. We demonstrate its role in cancer cell and tumor growth and provide evidence for its association with worse survival in a subset of breast cancer patients. Importantly, our genome-wide analyses have revealed novel insights into the function of this lncRNA, demonstrating how it changes the microRNA (miRNA) landscape leading to genome-wide mRNA expression regulation. Our study suggests that *PART1* represents an attractive target for the treatment of triple-negative breast cancers.

**Abstract:**

Triple-negative breast cancers (TNBCs) are aggressive, lack targeted therapies and are enriched in cancer stem cells (CSCs). Novel therapies which target CSCs within these tumors would likely lead to improved outcomes for TNBC patients. Long non-coding RNAs (lncRNAs) are potential therapeutic targets for TNBC and CSCs. We demonstrate that lncRNA prostate androgen regulated transcript 1 (*PART1*) is enriched in TNBCs and in Aldefluor^high^ CSCs, and is associated with worse outcomes among basal-like breast cancer patients. Although *PART1* is androgen inducible in breast cancer cells, analysis of patient tumors indicates its androgen regulation has minimal clinical impact. Knockdown of *PART1* in TNBC cell lines and a patient-derived xenograft decreased cell proliferation, migration, tumor growth, and mammosphere formation potential. Transcriptome analyses revealed that the lncRNA affects expression of hundreds of genes (e.g., myosin-Va, *MYO5A*; zinc fingers and homeoboxes protein 2, *ZHX2*). MiRNA 4.0 GeneChip and TaqMan assays identified multiple miRNAs that are regulated by cytoplasmic *PART1*, including *miR-190a-3p*, *miR-937-5p*, *miR-22-5p*, *miR-30b-3p*, and *miR-6870-5p*. We confirmed the novel interaction between *PART1* and *miR-937-5p*. In general, miRNAs altered by *PART1* were less abundant than *PART1*, potentially leading to cell line-specific effects in terms miRNA-*PART1* interactions and gene regulation. Together, the altered miRNA landscape induced by *PART1* explains most of the protein-coding gene regulation changes (e.g., *MYO5A*) induced by *PART1* in TNBC.

## 1. Introduction

Accumulating evidence supports a role for long non-coding RNAs (lncRNAs) in the development and progression of cancer and response to therapy. Defined as non-coding transcripts greater than 200 bases in length, the number of lncRNAs identified in humans has surpassed the number of coding genes; however, the function of only a fraction is known. Functionally characterized lncRNAs have been shown to affect gene expression by acting as activators or decoys for transcription factors, recruiters of chromatin-modifying complexes, miRNA sponges, and scaffolds of molecular complexes [1,2,3]. LncRNAs often display tissue-specific expression patterns and selective expression in certain cancers, making them attractive therapeutic targets [4]. In breast cancer for example, lncRNAs *nuclear paraspeckle assembly transcript 1* (*NEAT1*), *metastasis-associated lung adenocarcinoma transcript 1* (*MALAT1*), and *non-coding RNA in the aldehyde dehydrogenase 1A pathway* (*NRAD1*), have all been described as oncogenic and dysregulated in breast cancer [5,6,7]. Furthermore, hundreds of lncRNAs (most of which are uncharacterized) are upregulated in triple-negative breast cancers (TNBCs) and represent an area of potential discovery in terms of functionally important players in the progression of this subtype [8].

TNBCs lack the hormone receptors required for endocrine therapy in the treatment of breast cancer [9,10]. Hence, TNBC lacks targeted therapies and is treated with chemotherapy, which contributes to worse outcomes for patients with this subtype. Gene expression analysis of breast tumors revealed five major subtypes; luminal A, luminal B, HER2 overexpressing, basal-like, and claudin-low. The majority of TNBCs are basal-like and claudin-low. TNBC/basal-like breast cancers also have higher percentages of cancer stem cells (CSCs) [11,12,13,14,15,16,17], which may contribute to the aggressiveness of these cancers. CSCs are highly tumorigenic, have stem-like qualities, are resistant to chemo- and radiotherapies, and are commonly defined by increased aldehyde dehydrogenase (ALDH) activity detected by the Aldefluor assay or CD44^high^/CD24^low^ cell surface marker expression [18]. Given the high abundance of CSCs within TNBC/basal-like breast cancer [11,12,13,14,15,16,17], novel therapies that target CSCs may better reduce the risk of relapse and improve patient outcomes.

One emerging class of therapies for treating TNBCs and CSCs are lncRNA antagonists. For example, targeting tumor-specific lncRNAs with modified antisense oligonucleotides (termed GapmeRs [19]) inhibited tumor progression, metastasis and increased response to other therapies [20,21]. Thus, pharmacological inhibition of lncRNAs may be an effective therapeutic strategy, especially with the FDA’s recent approval of antisense oligonucleotide-based therapies for the treatment of neurodegenerative disorders [22].

One lncRNA of potential interest is *prostate androgen regulated transcript 1* (*PART1*). Discovered in 2000, *PART1* is expressed in prostate tissue and is responsive to androgens in prostate cancer cell lines [23]. *PART1* promotes prostate cancer cell proliferation and inhibits cell apoptosis, and is associated with more advanced disease and poorer survival among prostate cancer patients [24]. *PART1* expression is also associated with worse outcomes and higher stage disease in non-small cell lung cancer [25] and gastric cancer progression [26]. In colorectal cancer, *PART1* is oncogenic through possibly acting as miRNA sponge of *miR-143* and regulating *DNA methyltransferase 3A* (*DNMT3A*) [27]. Similarly, *PART1* promotes resistance of epidermal growth factor receptor inhibitor gefitinib in esophageal cancer by possibly acting as a miRNA sponge of miR-129, leading to increased levels of anti-apoptotic *BCL2* [28]. In contrast to these studies, *PART1* is part of lncRNA signature that predicts lower risk disease in glioblastoma [29] and has been implicated as a tumor suppressor in glioma cells [30]. To date, there has been limited analysis of the role of *PART1* in breast cancer with only one study reporting pro-oncogenic activities associated with the lncRNA in MCF7 and BT20 breast cancer cells [31].

Herein, we characterize the expression and function of *PART1* in breast cancer using multiple “omics” approaches. First, *PART1* expression in patient tumors is predominately in the TNBC/basal-like subtype and is associated with worse outcomes among basal-like breast cancer patients. We report its androgen inducibility in breast cancer cell lines; however, based on gene expression analysis of patient tumors and cells, androgen regulation of *PART1* appears to have a minimal clinical impact in breast cancer. We demonstrate that *PART1* is oncogenic in TNBC, limiting cell proliferation and decreasing migration when it is knocked down. Furthermore, *PART1* expression is associated with stemness gene expression and CSC markers in breast cancer patient tumors, Aldefluor^high^ sorted CSC populations, and mammosphere formation potential. Finally, we performed the first unbiased genome-wide analyses of *PART1* function, revealing that the cytoplasmic lncRNA regulates expression of hundreds of genes in TNBCs. Most of the changes in protein-coding gene expression are connected to corresponding changes in miRNAs that are predicted to either bind to *PART1* or are increased by *PART1*. Additionally, we confirmed a new miRNA interaction between *PART1* and *miR-937-5p*.

## 2. Materials and Methods

### 2.1. Cell Line Models and Tumor Studies

All the breast cancer cell lines were obtained from the American Type Culture Collection (ATCC, Manassas, VA, USA), except for SUM149 and SUM1315 cells, which were obtained from BioIVT (previously Asterandm Westbury, NY, USA). The cell lines were cultured as per ATCC and BioIVT recommendations. The TNBC patient-derived xenograft (PDX) 7482 (originated from a grade 3, stage 2 primary tumor, breast carcinoma) was obtained as a low-passage cryopreserved tumor piece from Dr. Michael T. Lewis and Lacey E. Dobralecki of the Patient-Derived Xenograft and Advanced In Vivo Models (PDX-AIM) Core of Baylor College of Medicine (Houston, TX, USA). All animal experiments have been conducted in accordance with the ethical standards and according to the Declaration of Helsinki and the Canadian Council on Animal Care (CCAC) standards and were approved under animal protocol 17-011.

For isolation of PDX tumor cells, the cryopreserved PDX was first revived by surgical implantation in the mammary fat pad of a NOD/SCID female mouse. The tumor expanded for 5 weeks and was then harvested from the euthanized mouse.

For tumor studies with HCC1806 cells, eight-week-old NOD/SCID female mice were injected with 10,000 HCC1806 shRNA control or *PART1* shRNA1 cells admixed 1:1 with matrigel-HC (ThermoFisher Scientific, Waltham, MA, USA) into the mammary fat pad (*n* = 7). Tumor volumes were quantified with caliper measurements (mm^3^, length × width × width/2). Mice were grasped, the tumor area was wetted to prevent the fur from obscuring the visibility of the tumor, and the tumor was measured in the longest dimension (length, l) and the second longest dimension (width, w). For the *PART1* shRNA1 group, two of the 7 mice did not have detectable tumors, so we scored them as 0 for tumor volume. However, when we performed a necropsy at termination, we detected tumors in all the mice (all 14 mice, 7n per group), hence the final tumor weights were determined for all 14 mice.

### 2.2. R1881 and D36 Treatment in Cell Culture

For studies assessing the effect of androgens on *PART1* induction, cell monolayers were cultured in phenol-red free media and charcoal-stripped FBS (ThermoFisher Scientific) and treated for 24 h with 10 nM R1881 (Sigma-Aldrich, Oakville, ON, Canada) and/or 40 µM D36 (Sigma-Aldrich). Total RNA was extracted and levels of *PART1* were assessed by reverse transcription-quantitative PCR (QPCR) as described below.

### 2.3. Total RNA Extraction and Reverse Transcription-Quantitative PCR

For all transcript expression analyses by QPCR, cells were collected in TRIzol and total RNA was purified using a PureLink RNA kit (Thermo Fisher Scientific) following the manufacturer’s instructions. Equal amounts of harvested RNA were reverse transcribed with the iScript cDNA Synthesis Kit (Bio-Rad, Saint Laurent, QC, Canada) as per the manufacturer’s instructions. QPCR was performed using SsoAdvanced Universal SYBR Supermix (Bio-Rad) and transcript-specific primers (primer sequences are listed in Appendix A) as per the manufacturer’s recommended protocol using a CFX96 Touch RealTime PCR Detection System (Bio-Rad). Primer efficiencies, determined by standard curves of diluted cDNA samples, were incorporated into the CFX Manager software (Bio-Rad). Gene expression of all samples was calculated relative to two or three reference genes as indicated in the figure legend and relative to the negative control (Appendix A).

### 2.4. Knockdown of PART1, Cell Proliferation, Migration, Apoptosis, Mammosphere Assays

Stable *PART1* short hairpin ribonucleic acid (shRNA) knockdown clones were generated in HCC1806 cells using two separate shRNA clones (shRNA 1, TAGTCGTAATTGAGTTCTGAC; shRNA2, AATAATGGGACATCACTTC, Dharmacon Inc., Lafayette, CO, USA) or scramble shRNA controls in lentiviral vectors pGipZ or pLKO.1. The lentiviral supernatants for transfection were generated in HEK293T cells using standard protocols. The lentiviral supernatants were applied to HCC1806 cells and clones were selected by adding 1.5 µg/mL puromycin containing media for two days and subsequently maintained in 0.25 µg/mL puromycin containing media. Knockdown of *PART1* was confirmed by QPCR as described above.

For transient in vitro *PART1* knockdown, 15 nM screening-grade modified antisense oligonucleotide GapmeRs (Qiagen, formerly Exiqon, Toronto, ON, Canada) were applied to cells (negative control GapmeR, LG00000002-DDA, AACACGTCTATACGC; *PART1* GapmeR 1, LG00211399-DDA, ATTCCAGATAAGTAGA; *PART1* GapmeR 2, LG00211400-DDA, GTGATTCCAGAATAAGT). GapmeRs were mixed with OptiMEM reduced serum media (Invitrogen ThermoFisher Scientific) and TransIT-BRCA transfection reagent (MJS Biolynk, Brockville, ON, Canada) and added to subconfluent HCC1806 or HCC1395 cells to a final treatment concentration of 15nM as per the manufacturer’s instructions. To quantify GapmeR-mediated decreases in *PART1* expression, QPCR analysis was performed on cells that were treated for 48 h and collected in TRIzol. Total RNA was purified using a PureLink RNA kit (as described above).

Effects on cell proliferation and apoptosis were measured by cell counting via trypan blue exclusion or flow cytometry analysis. Flow cytometry was performed using annexin-V conjugated to APC (Invitrogen Thermo Fisher Scientific) and *7*-aminoactinomycin D (7AAD, Biolegend, San Diego, CA, USA) using a FACSCalibur or FACSCelesta (BD BioSciences, San Jose, CA, USA) and FCSExpress flow cytometry analysis software (version 4, De Novo Software, Pasadena, CA, USA).

The motility of cells upon *PART1* knockdown was assessed by a gap closure assay. Cells were seeded at 20,000 cells per well in 2-well culture inserts placed in a 6-well plate. After overnight incubation, the cells were treated with 10 μg/mL mitomycin (a cell proliferation inhibitor) and incubated at 37 °C in 5% CO_2_ in a humidified incubator for 2 h. The cells were then treated with GapmeRs as described above for 24h. The inserts were then removed, and the first microscopic image was captured with a conventional 10× phase-contrast objective lens. A second microscopic image was captured 24 h later. The number of migrated cells in the gap was quantified in the second image.

To assess the effects of *PART1* knockdown on mammosphere formation potential, 3000 HCC1806 cells, 4000 HCC1395 cells or 5000 PDX 7482 cells were seeded in complete MammoCult media (Stemcell Technologies, Vancouver, BC, Canada) in technical triplicates in 24-well ultralow adherence plates (ThermoFisher Scientific). The PDX 7482 cells were obtained from tumors which had been expanded in a NOD/SCID mouse as described above. The expanded tumors were minced, treated with 225U/mL collagenase 3 (Bioshop Canada Inc., Burilington, ON, Canada) for 1 h at 37 °C with rotation, and strained through a 70 μM filter. Next, the red blood cells were lysed, washed in PBS and counted with a hemocytometer by trypan blue exclusion. Two hours post seeding, cells were treated with 15 nM GapmeRs as described above. All resulting spheres greater than 50 μm [32,33] (defined using the integrated software of an AE31E microscope (Motic, Richmond, BC, Canada), were counted 5 days later for HCC1806 and HCC1395 cells or 14 days later for PDX 7482 cells.

### 2.5. Aldefluor Analysis and Sorting

Aldefluor^high^ and Aldefluor^low^ populations of PDX 7482 were isolated using FACSAria (BD Biosciences) based on Aldefluor activity (Aldefluor assay kit, StemCell Technologies) performed as per the manufacturer’s instructions and as previously described [34,35]. Single cell suspensions of PDX 7482 were generated as described above. To remove dead cells and lineage cells of mouse origin, the cells were stained with 7-AAD (Biolegend) and allophycocyanin (APC) conjugated anti-H2Kd antibody (Biolegend), respectively. The pan-ALDH inhibitor diethylaminobenzaldehyde (DEAB) was added to a sample to verify that an Aldefluor^high^ population of cells had been identified. RNA extraction and QPCR analysis as described above was performed on the sorted Aldefluor^low^ and Aldefluor^high^ cell populations.

To assess the effect of *PART1* knockdown on the percentage of Aldefluor positive cells, we performed the Aldefluor assay as described above but on cells that had been treated with GapmeRs for 48 h prior to collecting cells and performing the assay.

### 2.6. Western Blotting

HCC1806 cells with *PART1* knockdown (through shRNA or GapmeR treatment) for 48h were lysed in RIPA buffer. 50 µg of the cytoplasmic lysate was loaded along with cleaved caspase-3 control cell extracts (#9664, Cell Signaling, New England Biolabs Ltd, Whitby, ON, Canada) in a Mini-PROTEAN TGX Stain-Free Precast Gel (Bio-Rad) and ran for 1 h at 100 V in Tris-Glycine-SDS buffer. The lysates were transferred onto PVDF membranes in a Transblot-Turbo Transfer system (Bio-Rad) and blocked in 5% BSA for 1h at room temperature. The membrane was incubated with cleaved caspase-3 (Asp175, 5A1E) rabbit monoclonal antibody (#9663, Cell Signaling, 1:1000 in 5% BSA) overnight at 4 °C followed by peroxidase affiniPure goat anti-rabbit IgG (H + L, #111-035-144, Jackson Immunoresearch, West Grove, PA, USA) antibody (1:1000 in 5% BSA) for 1 h at room temperature. The chemiluminescence was imaged with the ChemiDoc imaging system (Bio-Rad). Subsequently, the blot was re-probed for actin (#13E5, Cell Signaling) and similarly imaged.

### 2.7. Subcellular Localization of PART1

The LncATLAS database was accessed to determine the relative concentration index (RCI) of *PART1* in the nuclear versus cytoplasmic compartments in a panel of cell lines calculated from RNA-seq [36]. We determined the relative amounts of *PART1* in the nucleus versus cytoplasm of HCC1806 cells by subcellular fractionation. HCC1806 cells were collected and lysed in cold hypertonic lysis buffer (10 mM Tris, 10 mM NaCl, 3 mM MgCl_2_, 0.3% NP-40 and 10% glycerol, pH 7.5, with RNAse inhibitor SUPERase-In, ThermoFisher Scientific) and pelleted by centrifugation. The supernatant (cytoplasmic fraction) was separated from the nuclear fraction pellet. The nuclear pellet was repeatedly washed with the cold hypertonic buffer. The cytoplasmic fraction was precipitated by sodium acetate/ethanol precipitation and pelleted by subsequent highspeed centrifugation. The RNA from both the nuclear and cytoplasmic fractions was extracted as described above. *PART1* levels were determined by QPCR and compared to nuclear lncRNA *NEAT1* and cytoplasmic lncRNA *DANCR* [19,37].

### 2.8. Dataset Analyses

Breast cancer patient clinical data and *PART1* stem gene co-expression data were extracted via cBioPortal from The Cancer Genome Atlas (TCGA) PanCancer Atlas and Cell 2015 (RNA-seq data) and the METABRIC (gene chip data) datasets [38,39,40]. Survival analysis was also completed using KMPlotter [41] and from data extracted from cBioportal (TCGA Cell 2015, PanCancer Atlas and METABIC datasets) [38,39,40]. RNA-seq expression of *PART1* and androgen receptor (AR) in the CCLE database was retrieved using the CCLE Broad Institute portal (portals.broadinstitute.org/ccle) [42].

### 2.9. Transcriptome Analysis

HCC1806 and HCC1395 cells were treated with negative control GapmeR, GapmeR#1 or GapmeR #2 for 48 h, collected in TRIzol reagent and RNA purified (as described above). The samples (*n* = 3) were sent to The Centre for Applied Genomics (TCAG, The Hospital for Sick Kids, Toronto, ON, Canada) for Affymetrix Human Gene 2.0 ST gene chip platform analysis (ThermoFisher Scientific). The data was processed with the Transcriptome Analysis Console (Affymetrix) to reveal differential gene expression. The raw data is deposited on the NCBI Gene Expression Omnibus (GEO) (GSE156114).

### 2.10. MiRNA Analyses

HCC1806 and HCC1395 cells were treated with negative control GapmeR, GapmeR #1 or GapmeR #2 for 48 h as described above. Total RNA was isolated using the mirVana™ miRNA Isolation Kit (ThermoFisher). Three biological replicates were sent to the TCAG for Affymetrix’s (Applied Bioscience) genechip miRNA 4.0 array analysis, which interrogated all miRNA sequences in miRBase Release 20. The data was processed with the Transcriptome Analysis Console (Affymetrix) to reveal differential gene expression. The raw data is deposited on GEO (GSE163569). Validation of the miRNA gene chip array and individual miRNA quantification was conducted on total RNA (purified using the mirVana miRNA isolation kit) with the pre-formulated primer from the TaqMan miRNA assays (for *RNU48*, *cel-miR-39*, *miR-129*, *miR-373-3p*, *miR-429*, *miR-635*) or TaqMan Advanced miRNA assays (for *miR-21-5p*, *miR-22-5p*, *miR-26b-5p*, *miR-30b-3p*, *miR-190a-3p*, *miR-937-5p*, *miR-6870-5p*, ThermoFisher Scientific). For the TaqMan miRNA assay, cDNA was synthesized from the total RNA using the pre-formulated reverse transcription primers and the reagents in the TaqMan microRNA reverse transcription kit. For the TaqMan miRNA advanced assays, from the total RNA, the specific mature miRNAs are extended in the 3′ end of the mature transcript through poly(A) addition, then lengthened in the 5′ end by adaptor ligation. The modified miRNAs then underwent universal reverse transcription followed by amplification to uniformly increase the amount of cDNA for all miRNAs (miR-Amp reaction). For both assays, TaqMan Fast Advanced Master Mix (ThermoFisher) was used for the QPCR. MiRNA levels detected using either the TaqMan miRNA assay and TaqMan miRNA advanced assay were calculated relative to the reference miRNAs (*miR-221* and *RNU48*, and *miR-21-5p* and *miR-2b6-5p*, respectively) and relative to the negative control.

We utilized the online tool LncBase v2 to identify in silico predicted miRNA target binding sequences on *PART1*, identified with the DIANA-microT algorithm [43]. TargetScan [44] was accessed to identify the predicted mRNA targets for miRNAs regulated by *PART1* in TNBC. The *PART1*-miRNA-mRNA network was visualized using the Cytoscape platform [45].

### 2.11. Luciferase Reporter Assay for PART1—miR-937-5p Interaction

Oligos specific to the wildtype (WT) *PART1* miRNA 937-5p binding region and the mutated version of the sequence (MUT) are listed in Appendix A. To make double stranded sequences for cloning, the oligos were admixed into oligo annealing buffer and heated to 90 °C for 3 min, followed by cooling to 37 °C for 15 min. The WT and MUT annealed oligos (ThermoFisher Scientific) were cloned into the multiple cloning site of the pmirGLO Dual-Luciferase miRNA Target Expression Vector (ThermoFisher Scientific, using SacI and XhoI restriction enzymes (New England Biolabs Ltd.). The confirmed vectors were co-transfected into HCC1806 cells with the pRLTK vector (Promega ThermoFisher Scientific), using TransIT-BRCA transfection reagent. 24 h later the mirVana miRNA negative control mimic or mimic-hsa-miR-937-5p (ThermoFisher Scientific) was transfected into the cells using TransIT-BRCA. The resulting firefly and renilla luciferase activity in the cells were measured 24 h later using the Dual-Glo^®^ Luciferase Assay System (ThermoFisher Scientifc) with a SpectraMax^®^ M3 Multi-Mode Microplate Reader (ThermoFisher Scientific). Binding of the mimic sequence to the luciferase reporter vector would inhibit production of luminescence.

### 2.12. Statistics

All statistical analyses were performed in the GraphPad Prism software (GraphPad Software, San Diego, CA, USA). In all cases where three or more groups are compared, a one-way or two-way ANOVA was performed (with Dunnett’s or Tukey’s multiple comparisons post-test as indicated in the figure legend). Comparisons between two groups were done using a two-tailed student’s *t*-test. For co-expression analyses, *p* values were determined by the cor.test() function with the method argument set to “spearman” in Rv4.2. Significant *p* values are indicated as follows in the figures: *p* < 0.05 = *, *p* < 0.01 = **, *p* < 0.001 = ***, *p* < 0.0001 = ****.

## 3. Results

### 3.1. PART1 Is Enriched in Basal-Like/Triple-Negative Breast Cancer Patient Tumors and Is Androgen Inducible in Breast Cancer Cells

To define the potential role of *PART1* in breast cancer, we first analyzed *PART1* expression in breast cancer patient tumors of different subtypes (TCGA PanCancer, Cell 2015 and METABRIC datasets), based on intrinsic molecular subtype (basal-like, claudin low, HER2, luminal A and B), and the lack of hormone receptor expression (i.e., TNBC) [46,47]. This revealed that *PART1* is most expressed in basal-like tumors (Figure 1A, which are predominately TNBCs [46,47]) and in TNBCs (Figure 1B). To assess if *PART1* expression followed the same subtype-specific trends in breast cancer cell lines, we analyzed *PART1* expression in a 57-breast cancer cell line panel (Figure 1C, CCLE dataset, RNA-seq) and in our 24-breast cell line panel by QPCR (Figure 1D). Interestingly, in the cell lines *PART1* expression was not enriched in the basal-like/TNBC cells (Figure 1C,D) as we had noted in patient tumors (Figure 1A,B).

We wondered if the higher *PART1* expression in non-TNBC cell lines is a result of cell culturing conditions (i.e., androgenic signaling molecules are present in the phenol red/FBS containing media). Given that *PART1* has been shown to be responsive to androgens in prostate cancer cells [23], we suspected that the androgens in the cell culture media were influencing *PART1* expression in breast cancer cells. Notably, in the breast cancer cell lines, *androgen receptor* (*AR*) is weakly, positively correlated with *PART1*, whereas in breast cancer patient tumors, *AR* is significantly negatively correlated with *PART1* (Figure 1E,F), potentially suggesting a cell culturing-dependent effect. *AR* expression is lowest in basal-like/TNBC cell lines (e.g., HCC1806 and HCC1395) and highest in luminal/estrogen receptor positive (ER+) cell lines (e.g., T47D, Figure 1G).

We similarly noted that basal-like breast cancer patient tumors had the lowest *AR* expression (Figure 1H). This suggests that in the higher *AR* expressing ER+ breast cancer cell lines cultured in androgen-containing media, *PART1* expression may be at least partially dependent on androgen signaling.

To test this hypothesis, we cultured TNBC HCC1806 cells and ER+ T47D cells in phenol-red free/charcoal-stripped FBS, with or without the addition of 10 nM synthetic androgen R1881. This resulted in a modest induction of *PART1* expression in basal-like HCC1806 cells (1.26-fold, Figure 1I), and a much more significant induction of *PART1* in T47D cells (1.73-fold, Figure 1I). This is consistent with the higher levels of *AR* in T47D cells versus HCC1806 cells (Figure 1G). Addition of *AR* antagonist D36 inhibited the induction of *PART1* by R1881, confirming the role of *AR* on *PART1* levels (Figure 1J). Therefore, *PART1* expression can be amplified by the presence of androgenic signaling molecules in *AR* expressing breast cancer cell lines (e.g., ER+ T47D cells). To assess the potential clinical relevance of *PART1*/AR signaling in breast cancer, we assessed the correlation of *PART1* expression with the androgen signaling gene panel (containing 10 genes, from cBioPortal) across breast cancer subtypes in two breast cancer patient tumor datasets (Appendix A). This failed to reveal strong correlations between androgen signaling genes and *PART1* in breast cancer patient tumors. Together this data leads us to conclude that in TNBC/basal-like breast cancer, where *PART1* expression is highest and most likely clinically relevant, androgens do not play a major role in inducing *PART1* expression.

### 3.2. PART1 Is Oncogenic in Triple-Negative Breast Cancer Cells

Given the predominant expression of *PART1* in TNBC patients (Figure 1A,B), we focused our functional analyses to TNBC cell lines. We prioritized adherent TNBC cell lines with the highest *PART1* expression (HCC1806 and HCC1395) for our assays. Since the androgen induction response (although significant) is minimal in TNBC cells (Figure 1I), and that there is no connection between androgen signaling and *PART1* in breast cancer patient tumors (Appendix A), we opted to not use charcoal-stripped FBS and phenol red-free media for the functional assays.

We were able to generate a modest stable knockdown of *PART1* in HCC1806 cells (Figure 2A). We observed that *PART1* knockdown decreased HCC1806 cell proliferation (Figure 2B). To assess the role of *PART1* in vivo, we injected HCC1806 scramble control and *PART1* knockdown cells into the mammary fat pads of several NOD/SCID mice and found that *PART1* knockdown significantly decreased tumor volumes and tumor weights (Figure 2C). Consistent with the in vitro and in vivo data, high *PART1* expression was generally associated with worse survival in basal-like breast cancers patients based on median cutoffs for high versus low expression (Figure 2D, Appendix A).

Given that antisense oligonucleotides (GapmeRs) can also be used to target lncRNAs, we treated HCC1806 and HCC1395 cells with *PART1*-specific GapmeRs. This resulted in decreased *PART1* expression in both cell lines (Figure 2E), and a corresponding decrease in cell proliferation (Figure 2F), and migratory capacity (Figure 2G). We detected some effects on apoptosis in the *PART1* shRNA knockdown clones, but none in the GapmeR-treated cells, nor detectable caspase 3 cleavage, leading us to conclude that *PART1* inhibition has minimal effects on apoptosis (Appendix A). Together, these results support the hypothesis that *PART1* is oncogenic in breast cancer. Furthermore, since *PART1*-mediated cell proliferation and migration can be reduced using *PART1*-specific antisense oligonucleotides (which can be applied therapeutically), *PART1* may represent a novel therapeutic target for TNBC.

### 3.3. PART1 Expression Is Associated with Stemness Gene Expression, Aldefluor^high^ Cancer Stem Cells, and Contributes to Mammosphere Formation Potential

Given that basal-like/TNBCs have higher proportions of tumor-initiating CSCs relative to other subtypes [11,12,13,14,15,16,17], we wondered if *PART1* expression is also associated with CSCs. We assessed the co-expression of *PART1* with CSC markers and stemness genes in breast cancer patient tumor datasets (Figure 3A,B, Appendix A), and noted significant correlations, especially in the basal-like subtype. Across both patient tumors datasets, *PART1* expression was significantly positively associated with expression of CSC marker *ALDH1A3*, the primary cause of the Aldefluor activity in breast CSC cells [34], and CSC-associated gene *integrin alpha 6* (*ITGA6*, also known as *CD49f*), which is regulated by ALDH1A3 [48].

We used the Aldefluor assay to interrogate the expression of *PART1* in Aldefluor^high^ sorted cells and in the TNBC patient-derived xenograft (PDX) model 7482. Notably, we have previously determined that the sorted Aldefluor^high^ cells of PDX7482 are more tumorigenic in mice than Aldefluor^low^ cells [5]. Hence the PDX has been validated as having CSC populations defined by high Aldefluor activity. *PART1* expression was enriched in the sorted Aldefluor^high^ populations (Figure 3C), in agreement with the patient gene expression data (Figure 3A,B).

We assessed if the GapmeR treatment affected the percentage of Aldefluor^high^ cells directly (Appendix A). We noted no significant changes on the percentages of Aldefluor^high^ cells, suggesting that treatment with *PART1* GapmeR does not selectively target Aldefluor^high^ cells. Breast CSCs/tumor initiating cells (TICs) have an increased capacity to form mammospheres in non-adherent cell culture conditions and sphere-forming potential is an accepted in vitro readout of tumor forming capacity and stemness [32,48,49]. Thus, we evaluated whether *PART1* inhibition affects the mammosphere-forming potential of TNBC PDX 7482, HCC1806, and HCC1395 cells. Treatment of the cells with anti-*PART1* GapmeRs decreased the mammosphere forming potential of PDX 7482, HCC1806, and HCC1395 cells (Figure 3C–E). Together these results suggest that *PART1* is an Aldefluor^high^/CSC-associated lncRNA, that when inhibited, impairs mammosphere forming potential, thereby marking it as a potential anti-breast CSC target.

### 3.4. PART1 Induces Gene Expression Changes in Triple-Negative Breast Cancer Cells and Is Predominately Cytoplasmic

We next wondered how *PART1* was mediating these effects in TNBC. Given the predominate role of lncRNAs in gene regulation [37], it is likely that *PART1* contributes to gene expression regulation in TNBC cells. The microarray gene chip transcriptome analyses revealed that *PART1* knockdown with GapmeRs altered expression of hundreds of genes in HCC1806 and HCC1395 cells (Figure 4A, Appendix A). The heatmap analysis revealed a high degree of gene expression overlap between the two GapmeRs within the two cell lines, with partial overlap across the cell lines (Figure 4A).

To validate the gene chip array results, we performed QPCR on genes significantly differentially expressed upon *PART1* knockdown in one or both cell lines (Figure 4B). This confirmed both common (e.g., *myosin-Va*, *MYO5A*; zinc fingers and *homeoboxes protein 2*, *ZHX2*) and distinct gene regulation by *PART1* across cell lines, or in only HCC1395 (e.g., *bicaudal C homolog 1*, *BICC1*) or HCC1806 (e.g., *serine/threonine-protein phosphatase 2A regulatory subunit B*, *PPP2R3A*) cells.

Consistent with the oncogenic effects of *PART1* in the TNBC cells, *PART1* knockdown downregulated *MYO5A*, *ZHX2* and *BICC1*, which have all been implicated in cancer progression [49,50,51,52]. In contrast, *PART1* knockdown upregulated *PPP2R3A*, which is a suspected tumor suppressor [53]. We detected minimal changes in gene expression of CSC markers, stemness genes and androgen signaling genes (Appendix A and Appendix A).

It is also notable that most of the transcript changes induced by *PART1* in the TNBC cells are in non-coding genes, including other lncRNAs, miRNAs and small nuclear RNAs (Figure 4C). Considering that the Human Gene 2.0 ST Array has more coverage of the coding genome, the transcript changes induced by *PART1* are over-represented among non-coding genes in the TNBC cells. Given that regulatory nature of non-coding RNAs, *PART1* may be indirectly affecting protein-coding gene expression by modulating these transcripts. This could explain the distinct *PART1*-mediated gene regulation in the cell lines, which would also depend upon the cell-line specific abundance of these non-coding RNAs.

Sub-cellular fractionization partially defines the mode of gene regulation of a lncRNA, where nuclear lncRNAs are often chromatin modifiers and cytoplasmic lncRNAs may act as miRNA sponges (i.e., competitive endogenous RNAs, ceRNAs) [37]. Thus, to determine how *PART1* may be affecting gene expression, we first assessed its cellular localization. The LncATLAS database provides relative cellular fraction concentrations of over 6000 lncRNAs (determined by RNA-seq) across a panel of cell lines, including one breast cancer cell line, ER+/PR+ MCF7 cells [36]. In comparison to highly nuclear lncRNA *NEAT1* and cytoplasmic lncRNA *DANCR*, *PART1* is predominately cytoplasmic in the cell lines assessed by LncATLAS (Figure 4D). To validate these findings in TNBC, we performed subcellular fractionation of HCC1806 cells followed by QPCR on the nuclear and cytoplasmic fractions. Similar to the LncATLAS data, we found that *PART1* is predominately cytoplasmic in the TNBC cells (Figure 4E). These localization results, in addition to the large portion of non-coding transcripts regulated by *PART1* in the gene array, indicate that *PART1* may interact with miRNAs to regulate gene expression in TNBC.

### 3.5. PART1 Alters the miRNA Landscape in Triple-Negative Breast Cancer Cells

Previous reports have shown that *PART1* acts as a ceRNA on seven miRNA targets; *miR-635* [54], *miR-129* [28], *miR-373-3p* [55], *miR-429* [55], *miR-150-5p* [56,57], *miR-143-3p* [27], and *miR-190a-3p* [30]. We used TargetScan [55] to identify the predicted mRNA targets for each of these seven miRNAs. We then compared these mRNA hits to our list of genes downregulated or upregulated by *PART1* knockdown in TNBC cells (thresholds of ±1.6-fold change, *p* value < 0.05, Appendix A) to determine which miRNAs are predicted to regulate the same mRNAs as *PART1*. Of note, *miR-190a-3p* had the greatest number of predicted mRNA targets among our gene lists (Appendix A). In contrast, *miR-150-5p* and *miR-143-3p* had only one predicted mRNA target within our gene list.

We therefore proceeded with quantifying the levels of the five miRNAs that had at least 4 predicted mRNA hits within our gene list (i.e., *miR-190a-3p*, *miR-635*, *miR-429*, *miR-129*, *miR-373-3p*). We isolated mature miRNAs from the TNBC cells using the miRVana kit and quantified the levels of the miRNAs with specific TaqMan miRNA assays. Only *miR-190a-3p* was increased upon *PART1* knockdown in TNBC cells; however, it was significant in only one knockdown in HCC1395 cells (Figure 5A). This suggests that there are likely additional mechanisms contributing to the *PART1*-mediated gene regulation we describe in Figure 4.

None of the prior studies that identified miRNA targets of *PART1* performed an “omics” approach to identify all potential *PART1*-miRNA interactions (i.e., they focused on characterizing individual miRNA interactions). The DIANA web-based tool LncBase v2 predicts that there are over 400 mature miRNAs that could theoretically be sponged by at least one *PART1* transcript (threshold set to 0.7, Appendix A). This is perhaps not surprising given that *PART1* encodes multiple transcripts, the largest one being over 5000 bases in length, giving significant opportunity for miRNA interactions. While the miRNA-binding prediction tool does not consider factors such as *PART1* and miRNA abundance, nor the secondary and tertiary structure of the lncRNA (which would physically exclude certain interactions), it does suggest that an omics-based approach is warranted for experimentally identifying miRNAs that may be regulated by *PART1*. We therefore performed gene chip 4.0 miRNA arrays, designed to interrogate all miRNA sequences in miRBase Release 20 on HCC1806 and HCC1395 cells (Appendix A).

The miRNA gene chip array analysis revealed that *PART1* knockdown both increased and decreased several mature miRNAs in the TNBC cells (Figure 5B; Appendix A, volcano plots). We identified a list of miRNAs potentially regulated by *PART1* in TNBC cells that had predicted mRNA targets within our gene lists of *PART1*-regulated mRNAs (Figure 5B, Appendix A). The heatmap analysis of the miRNAs revealed a high degree of overlap between the two GapmeR-treated samples with partial overlap across the two cell lines (Figure 5B). We noted that miRNAs that were regulated by *PART1* with at least a 1.3-fold change were mostly less abundant than *PART1*, and generally between a miRNA:*PART1* abundance ratio of 0.25 to 1 (Figure 5C, Appendix A). This suggests that miRNA:*PART1* abundance is a cell line-specific factor in determining the effect of the lncRNA on a miRNA in a cell, regardless of its potential for interaction (i.e., sequence complementarity). Of note, the miRNAs that were previously described as being sponged by *PART1* in other tissues and cancers were lowly expressed in the TNBC cells (Appendix A). This could explain why we did not detect the regulation of *miR-635*, *miR-429*, *miR-129*, *miR-373-3p* by *PART1*. Perhaps most surprising was the number of miRNAs decreased upon *PART1* knockdown. Although less commonly described, there are reports of lncRNAs increasing miRNAs (e.g., lncRNA transcripts are processed into mature miRNAs) [58,59], indicating that lncRNAs can also modulate gene expression by inducing miRNAs.

We confirmed the miRNA array results with QPCR using TaqMan advanced miRNA assays on *PART1*-regulated miRNAs with at least 15 predicted mRNA targets in HCC1806 cells or 9 predicated mRNA targets in HCC1395 cells among our *PART1*-regulated gene lists (upregulated *miR-937-5p* upon *PART1* knockdown; downregulated *miR-22-5p*, *miR-30b-3p*, and *miR-6870-5p* upon *PART1* knockdown; Figure 5D). The miRNAs decreased upon *PART1* knockdown (*miR-22-5p*, *miR-30b-3p*, and *miR-6870-5p*) are not predicted to bind *PART1*. Furthermore, sequence analysis of *PART1* suggests that *miR-22-5p*, *miR-30b-3p*, and *miR-6870-5p* are not direct products of *PART1*, indicating they may be regulated by an indirect mechanism. In contrast, both *miR-937-5p and miR-190a-3p*, which are increased upon *PART1* knockdown, have predicted binding interactions with *PART1* transcripts (Figure 6A, Appendix A), suggesting that these miRNAs are sponged by *PART1*.

In terms of being sponged by *PART1*, the interaction between *PART1* and *miR-190a-3p* has already been demonstrated [30]; however, the interaction between *miR-937-5p* and *PART1* remains to be experimentally confirmed. As such, we performed a miRNA luciferase reporter assay where the target sequence (or the mutated version) is cloned into the miRNA reporter vector downstream of the luciferase reporter open reading frame [60,61]. The constructs, along with *miR-937-5p* mimic or negative control mimic, were transfected into HCC1806 cells. A target sequence interaction with *miR-937-5p* would result in decreased luminescence in the assay. We only observed a significant decrease in luminescence upon miR-937-5p treatment with the wildtype target sequence, confirming the novel interaction between the *PART1* sequence and miR-937-5p (Figure 6B).

To gain an appreciation for the full scope of the effects of *PART1*/miRNA-mediated gene regulation, we used TargetScan [55] to assess which of the mRNAs significantly regulated by *PART1* could interact with the TaqMan assay-validated miRNAs in HCC1806 or HCC1395 cells (Figure 5A,D). This revealed that most of the *PART1*-regulated mRNAs (protein-coding genes that had predicted interactions with the TaqMan assay confirmed-regulated miRNAs) had common hits between the miRNAs (Figure 6C, Appendix A). Together, these data suggest that an altered miRNA landscape by *PART1* affects 60% and 64% of the protein-coding genes regulated by *PART1* in HCC1806 and HCC1395 cells, respectively (Figure 6D). In the TNBC cells, *PART1* is at the center of a miRNA-mRNA network (Figure 6E). Novel interactions include *PART1*-*miR-937-5p*-*MYO5A*. Notably, we did not specify that upregulated miRNAs be paired with downregulated genes (and vice-versa) in the network, since miRNAs have been reported to induce mRNA transcripts, in contrast to their more commonly described translation inhibition and transcript decay effects [62]. The remaining gene expression changes induced by *PART1* that are not directly connected to this network (40% of protein-coding genes in HCC1806 cells and 36% in HCC1395 cells) are hence independent of miRNA regulation changes.

## 4. Discussion

Next-generation sequencing technologies have provided an alternative glimpse into the mammalian genome, revealing that most genomic products are transcribed into non-coding RNAs such as lncRNAs. Further evaluation of lncRNAs indicated that many are dysregulated in breast cancer, resulting in repercussions for breast cancer cell proliferation, tumor growth and metastasis [63]. For this reason, we explored phenotypic and functional characteristics of lncRNA *PART1* in breast cancer, which has been previously implicated in both oncogenic and tumor suppressive function in other cancer types including prostate, esophageal, lung, colorectal and glioblastoma. In esophageal and lung cancers, *PART1* is oncogenic [24,25,28,64]. Some evidence suggests that *PART1* may play a similar oncogenic role in breast cancer [31].

We used the METABRIC, Cell 2015, and PanCancer Atlas breast cancer datasets (extracted via cBioPortal) and KMplotter to assess *PART1* expression in different breast cancer subtypes and its association with clinical data. We show that *PART1* is most enriched in basal-like and TNBC patient tumors, although this did not appear to be due to increased *AR* signaling. There is likely another AR-independent regulation mechanism which leads to increased *PART1* in basal-like/TNBC patient tumors.

To clarify the potential role of *PART1* in breast cancer, we studied the consequence of its knockdown and inhibition in the TNBC cell lines and a PDX, assessed its expression correlations with breast cancer patient survival, and evaluated its association with CSC populations. More specifically, we revealed associations with *ALDH1A3* and Aldefluor^high^ CSC populations. This is an important distinction, since breast CSC populations can be defined by Aldefluor activity or CD44^high^/CD24^low^ cell surface marker expression, and these populations have distinctive features with only partial overlap [65]. Together, our analyses uniformly suggest that *PART1* exerts pro-survival/oncogenic effects in the cancer cells, including within CSC populations, likely due to increased *PART1* expression within these populations. Further, our use of *PART1*-specific antisense oligonucleotides suggests that the lncRNA could be targeted in the treatment of breast cancer and targeting of CSCs. In the future, experiments where *PART1* transcripts are overexpressed, resulting in increased mammosphere forming potential, proliferation and migration, would substantiate these data further.

In terms of function, the cumulative published data suggests that *PART1* regulates gene expression by acting as a miRNA sponge, although these studies were all specifically characterizing a single *PART1*/miRNA/mRNA axis. In colorectal cancer, *PART1* promotes tumor growth by acting as a ceRNA, sponging miR-143 [27]. Similarly, *PART1* was shown to promote malignant progression of colorectal cancer through the *miR-150-5p*/*LRG1* axis [56] and by sponging *miR-150-5p* to up-regulate the expression of *CTNNB1* and activate Wnt/β-catenin signaling [57]. Conversely, *PART1* has tumor suppressive function in glioma by sponging *miR-190a-3p*, leading to upregulation *miR-190a-3p* target *PTEN* (a tumor suppressor), which subsequently inactivates the PI3K/AKT pathway in glioma cell lines [30]. Our fractionation experiment suggests that *PART1* is predominately located in the cytoplasm in TNBC cells, which is consistent with the function of a miRNA sponge. Future analysis by fluorescence in situ hybridization (FISH) would provide further proof and can, in principle, determine absolute numbers of *PART1* molecules in the cellular compartments [64,66].

Based on its miRNA sponging activity, it is unsurprising that *PART1* can have both oncogenic and tumor suppressive effects. Depending on the function and abundance of the mRNAs targeted by the miRNAs, sponging of miRNAs by *PART1* can result oncogenic or tumor suppressive effects. For example, we found evidence of regulation of *miR-190a-3p* by *PART1*, but no subsequent effects on PTEN transcript levels, as was reported in the aforementioned glioma study [30]. We noted that PTEN is greater than 10-fold more abundant than *miR-190-3p* in HCC1806 and HCC1395 cells, which could explain why we failed to detect a corresponding effect on PTEN expression. Indeed, the abundance of mRNAs targeted by the miRNAs is a critical factor when considering the potential effects of a miRNA on mRNAs levels [67].

Our arrays and QPCR analyses revealed that *PART1* has genome-wide effects on gene expression in TNBC. *PART1* knockdown resulted in downregulation of cancer promoting genes like *BICC1*, *MYO5A* and *ZHX2*. MYO5A is an actin-dependent motor protein contributing to organelle transport. It is elevated in metastatic colorectal cancer and promotes migration and metastasis of lung, breast, and colon cancer cell lines [68]. MYO5A also promotes anchorage-independent growth, invasion and migration in melanoma [50]. The corresponding reduction in motility/migration upon *PART1* inhibition is consistent with the reduction *MYO5A* levels. Putative RNA-binding protein BICC1 contributes to cell viability and is associated with poor prognosis among patients with oral [69] and gastric cancers [49], consistent with proposed oncogenic role of BICC1. Finally, ZHX2 is a transcriptional repressor which promotes clear cell renal cell carcinoma soft-agar and tumor growth [52]. In contrast, *PART1* knockdown resulted in the upregulation of suspected tumor suppressor *PPP2R3A*, which encodes a regulatory subunit of protein phosphatase 2. *PPP2R3A* is highly methylated T- and B-acute lymphoblastic leukemia [70] and colon cancers [53], resulting in the silencing of this gene in these cancer. Thus, *PART1* may be mediating its oncogenic effects through regulation of these genes.

Using target prediction tools, we found that many of the protein-coding gene expression changes induced by *PART1* are linked to changes in miRNAs. In addition to known effects on *miR-190-3p*, we report novel *PART1*-mediated regulation of *miR-937-5p*, *miR-22-5p*, and *miR-30b-3p* in TNBC; all of which have been implicated in cancer progression in prior studies [71,72,73,74,75]. Additionally, we found that *PART1* regulates *miR-6870-5p*; however, in contrast to the miRNAs listed above, *miR-6870-5p* has not been studied previously and its function remains uncharacterized.

Our work demonstrates that *PART1* decreases and increases miRNAs; however, only some of the changes in miRNAs could be explained by *PART1*-mediated binding/sponging and none appear directly generated from *PART1*. This suggests that *PART1* is altering the miRNA landscape (at least in part) by indirect mechanisms. Intriguingly, a large portion of the *PART1*-induced gene expression changes were within the non-coding genome. The changes in expression of non-coding genes suggests indirect *PART1*-dependent mechanisms which could lead to changes in mature miRNAs and mRNA levels. It would be of interest to perform similar genome-wide analyses on the effects of *PART1* in other cancers and tissues to determine if the effects are generally true of *PART1*, or if they are specific to TNBC. Finally, we have not determined the effect of *PART1* on the proteome, which could mediate at least part of the *PART1* pro-oncogenic effects through interactions with proteins (e.g., by acting as a molecular scaffold). Hence, while we have advanced the knowledge of the role of this lncRNA in breast cancer, much remains to be studied to fully understand its function. Furthermore, it still needs to be determined if antisense oligonucleotides against *PART1* therapeutically reduce TNBC tumors in mice.

## 5. Conclusions

In recent years, increasing numbers of lncRNAs have been identified as playing important roles in breast cancer progression and some of these have been specifically associated within the CSCs populations of breast cancers. With completion of this study, we now add *PART1* to a growing shortlist of lncRNAs that are enriched in TNBC and breast CSCs. The targeting of *PART1* with antisense oligonucleotides and resulting negative impact on breast cancer cells suggest that *PART1* could be targeted in the treatment of TNBC. Mechanistically, our analyses go beyond characterizing a single mRNA and miRNA-*PART1* interaction (e.g., *miR-937-5p*-*PART1*) to genome-wide analyses of mRNA transcript and miRNA changes. This reveals new information regarding the many potential effects that the lncRNA has on miRNAs, which are only partly explained by sponging. Genomic analyses on *PART1* in other cancers will likely reveal similar effects; however, the miRNAs and mRNAs that are regulated by *PART1* will differ and depend upon the cellular context.

## Figures and Tables

**Figure 1 cancers-13-02644-f001:**
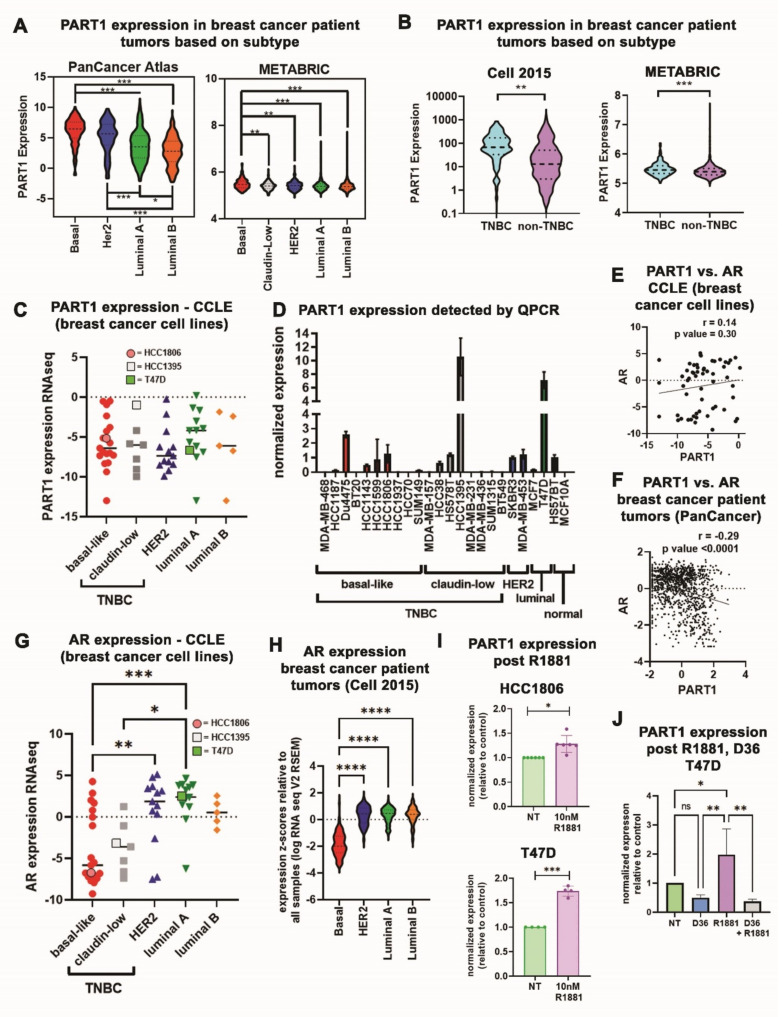
*PART1* expression is enriched in TNBC patient tumors and is induced by androgenic molecules in breast cancer cells. (**A**,**B**) *PART1* expression in breast cancer patient tumor cohorts based on molecular intrinsic subtypes, PAM50 or claudin-low (**A**) or TNBC status (**B**). Gene expression information was extracted from the TCGA PanCancer Atlas and Cell 2015 (RNA-seq) and the METABRIC (microarray) datasets via cBioPortal (1 May 2020). Significance comparing TNBC versus non-TNBC groups was determined using an unpaired two-tailed *t*-test, while comparisons of multiple groups determined with a one-way ANOVA followed by Tukey’s multiple comparison post-test. (**C**) Expression of *PART1* in 57 breast cancer cell lines (RNA-seq) was extracted from the CCLE and grouped based on molecular intrinsic subtypes, significance assessed with a one-way ANOVA followed by Tukey’s multiple comparison post-test (no comparison was significant). (**D**) Expression of *PART1* (QPCR) in 22 different breast cancer cell lines and two normal immortalized breast cell lines. Expression is relative to reference genes *ADP-ribosylation factor 1* (*ARF1*) and *pumilio RNA binding family member 1* (*PUM1*) (*n* = 4). Error bars represent standard error of the mean (SEM). (**E**) Expression of *PART1* versus *AR* in 57 breast cancer cell lines (RNA-seq) was extracted from the CCLE. The correlation (*r*) and *p*-value were determined by Pearson’s correlation coefficient. (**F**) Expression of *PART1* versus *AR* in breast cancer patient tumors from the Cell 2015 (RNA-seq) dataset was extracted via cBioPortal and the correlation (*r*) and *p*-value determine by Pearson’s correlation coefficient. (**G**) Expression of *AR* in 57 breast cancer cell lines (RNA-seq) was extracted from the CCLE and grouped based on molecular intrinsic subtypes, significance assessed with a one-way ANOVA followed by Tukey’s multiple comparisons post-test. (**H**) Expression of *AR* in breast cancer patient tumors based on molecular intrinsic subtypes was extracted from the Cell 2015 (RNA-seq) dataset via cBioPortal and a one-way ANOVA followed by Tukey’s multiple comparisons post-test was performed to determine significance. (**I**) The effect of R1881 synthetic androgen on *PART1* expression in HCC1806 and T47D cells was assessed by QPCR and is reported relative to reference genes (*PUM1* and *ARF1*) and control no treatment cells (*n* = 4–6). Significance was determined by a paired two-tailed *t*-test (error bars represent standard deviation, SD). (**J**) The effect of R1881 and *AR* antagonist D36 on *PART1* expression in T47D cells was assessed by QPCR and is reported relative to reference genes (*PUM1*, *ARF1*, and *beta-2 microglobulin* (*B2M*)) and control no treatment cells (*n* = 4). Significance was determined by two-way ANOVA followed by Tukey’s multiple comparisons post-test. Significant *p* values are indicated as follows in the figures: *p* < 0.05 = *, *p* < 0.01 = **, *p* < 0.001 = ***, *p* < 0.0001 = ****. Non-significant *p* values are either indicated as ns, or not noted.

**Figure 2 cancers-13-02644-f002:**
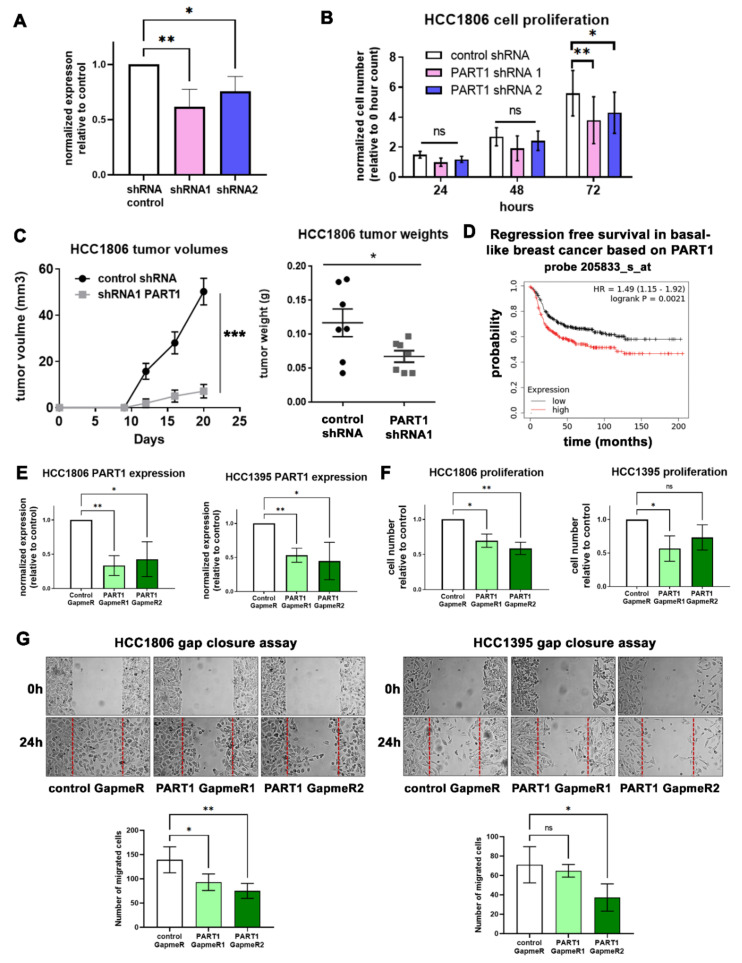
*PART1* expression promotes proliferation, migration, and tumor growth in TNBC cells. (**A**) *PART1* expression (QPCR) following shRNA-induced knockdown in HCC1806 cells (*n* = 4). Expression is shown normalized to reference genes *ARF1* and *PUM1* and control cells. Significance was determined by a one-way ANOVA, followed by Dunnett’s post-test for multiple comparisons). Error bars represent SD. (**B**) The effect of *PART1* knockdown on cell proliferation was quantified by counting the relative number of viable cells after 24, 48, and 72 h, using a trypan blue exclusion assay (*n* = 5, significance determined by a one-way ANOVA, followed by Dunnett’s post-test for multiple comparisons). Error bars represent SEM. (**C**) NOD/SCID mice were injected with either 10,000 HCC1806 scramble control shRNA clones or HCC1806 *PART1* shRNA 1 clone cells (*n* = 7). Tumor volumes were determined with caliper measurements (l × w^2^/2) and final tumor weights were determined at termination (significance determined by an unpaired *t*-test). Error bars represent SEM. (**D**) Kaplan-Meier survival curves generated by KMplotter. Survival was compared between high vs. low *PART1* (probe 205833_s_at) expression groups (where patients were stratified by median expression) in basal-like breast cancer (HR = hazard ratio). (**E**) QPCR analysis of *PART1* expression following *PART1*-specific GapmeR-mediated knockdown (GapmeR #1 and #2) relative to control GapmeR and reference genes in HCC1806 (ARF1 and PUM1) and in HCC1395 cells (*glyceraldehyde-3-phosphate dehydrogenase*, *GAPDH*; *beta-2 microglobulin*, *B2M*) (*n* = 4, error bars represent SD). (**F**) The effect of GapmeR-mediated *PART1* inhibition on cell proliferation was assessed by counting the relative number of viable cells 2 days after treatment using a trypan blue exclusion assay (*n* = 4, error bars represent SD). (**G**) The effect of GapmeR-mediated *PART1* inhibition on cell migration was assessed by gap closure assay (*n* = 4). Significance was determined by one-way ANOVA, followed by Dunnett’s post-test for multiple comparisons. Significant *p* values are indicated as follows in the figures: *p* < 0.05 = *, *p* < 0.01 = **. Non-significant *p* values are either indicated as ns, or not noted.

**Figure 3 cancers-13-02644-f003:**
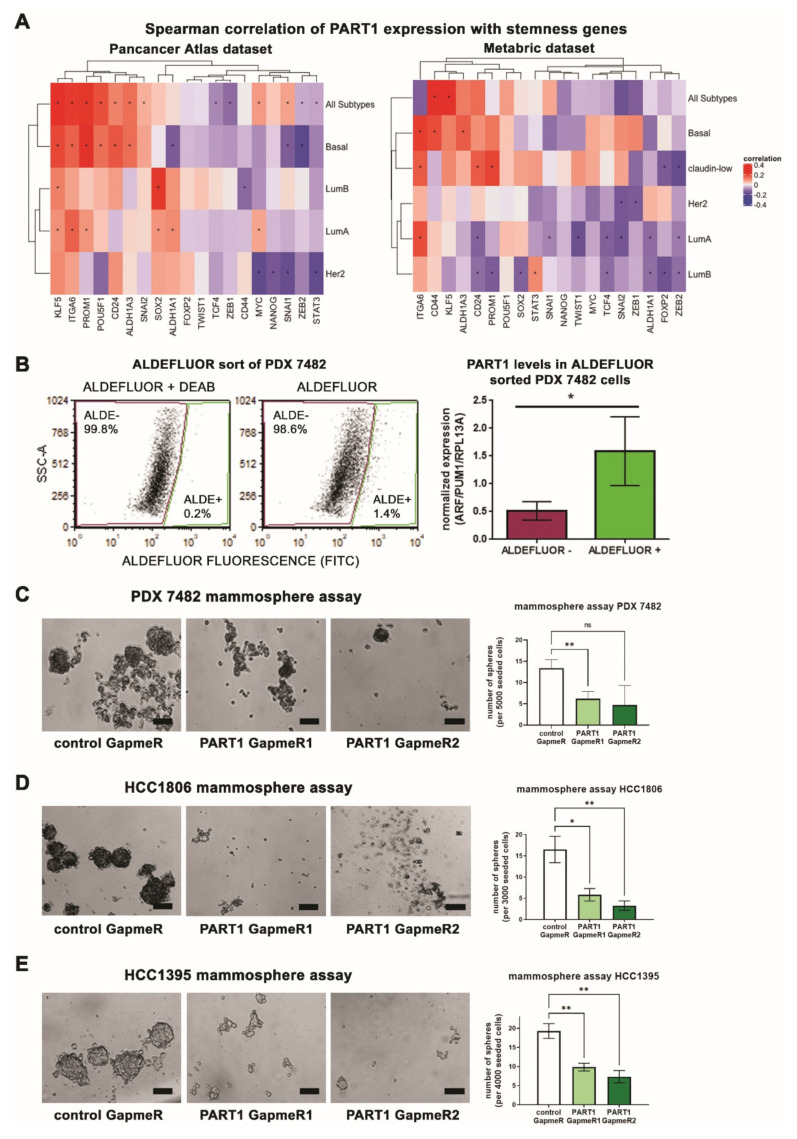
*PART1* is associated with Aldefluor^high^ breast CSC populations and its knockdown inhibits mammosphere formation in TNBC. (**A**) Spearman correlations of *PART1* expression with expression of stemness and CSC-associated genes in breast cancer patient tumors in two datasets (TCGA PanCancer Atlas and METABRIC). *p* values were determined by the cor.test() function with the method argument set to Spearman in Rv4.2. * indicates *p* value < 0.05. (**B**) Representative flow cytometry plots of the Aldefluor assay completed on PDX 7482 cells. The Aldefluor^high^ (ALDE+) and Aldefluor^low^ (ALDE-) were sorted. One sample had DEAB (an ALDH inhibitor) to ensure proper identification of the Aldefluor^high^ population. *PART1* expression in the sorted populations was determined by QPCR and made relative to the Aldefluor^low^ expression and normalized to reference genes (*n* = 3, significance was determined using a student’s *t*-test, error bars represent SD). (**C**–**E**) PDX 7482 (**C**), HCC1806 (**D**), and HCC1395 cells (**E**) were treated with 15 nM GapmeR in technical triplicates (negative control or *PART1*-specific GapmeR #1 and 2) and seeded at 5000 cells/well (PDX 7482, *n* = 3), 3000 cells/well (HCC1806, *n* = 4), or 4000 cells/well (HCC1395, *n* = 4) in ultra-low adherence plates. The average number of resulting spheres greater than 50 μm (the length of the scale bar) in diameter per well were counted (representative images are shown). Significance was determined using a one-way ANOVA, followed by Dunnett’s post-test for multiple comparisons (error bars represent SEM). Significant *p* values are indicated as follows in the figures: *p* < 0.05 = *, *p* < 0.01 = **. Non-significant *p* values are either indicated as ns, or not noted.

**Figure 4 cancers-13-02644-f004:**
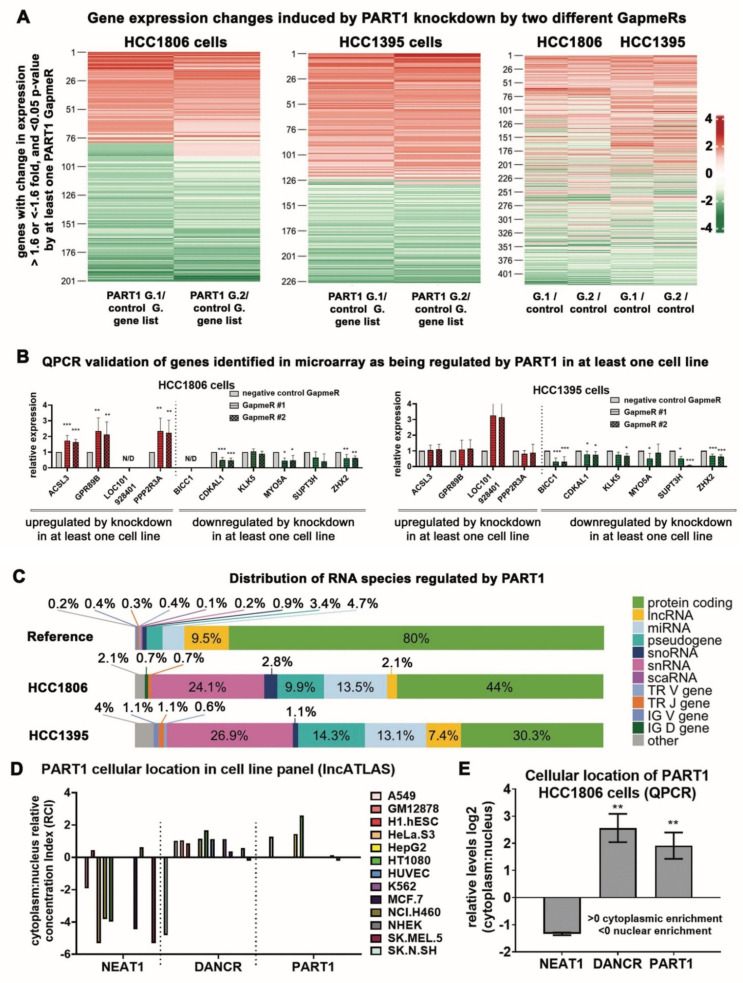
*PART1* induces gene expression changes in HCC1806 and HCC1395 TNBC cells and is cytoplasmic. (**A**) Transcriptome changes induced by *PART1* knockdown (Gapmer1 (G.1) or Gapmer2 (G.2) versus control GapmeR (control G.) were quantified in HCC1806 and HCC1395 cells using the Affymetrix Human Gene 2.0 ST microarray platform (*n* = 3). The heatmaps show genes with an expression fold change >1.6 or <−1.6 and a *p*-value < 0.05 induced by at least one *PART1*-specific GapmeR. (**B**) QPCR validation of some genes identified as upregulated by *PART1* knockdown (green bars) or downregulated (red bars) by *PART1* in (**A**) (*n* = 4–7, significance determined by one-way ANOVA, followed by Dunnett’s post-test for multiple comparisons). Error bars represent SD. Expression is normalized relative to the negative control and to reference genes *PUM1* and *ARF1*. (**C**) The portion of genes (protein coding and non-coding (lncRNA, miRNA, snRNA, pseudogene, misc RNA, snoRNA) covered by the microarray (top) and the portion of genes regulated by *PART1* that are in HCC1806 (middle) and HCC1395 (bottom) cells. (**D**) The LncATLAS [36] database was accessed to determine the relative concentration index (RCI) of *PART1* in the nuclear versus cytoplasmic compartments in a panel of cell lines by RNA-seq. Well-established nuclear-localized lncRNA *NEAT1* and cytoplasmic-localized lncRNA *DANCR* are included for comparison. (**E**) QPCR analysis of lncRNAs *DANCR*, *NEAT1*, and *PART1* abundance in nuclear and cytoplasmic fractions of HCC1806 cells. Relative expression versus *GAPDH* is shown (*n* = 3, significance was determined using student’s *t*-test). Significant *p* values are indicated as follows in the figures: *p* < 0.05 = *, *p* < 0.01 = **, *p* < 0.001 = ***. N/D signifies not detected.

**Figure 5 cancers-13-02644-f005:**
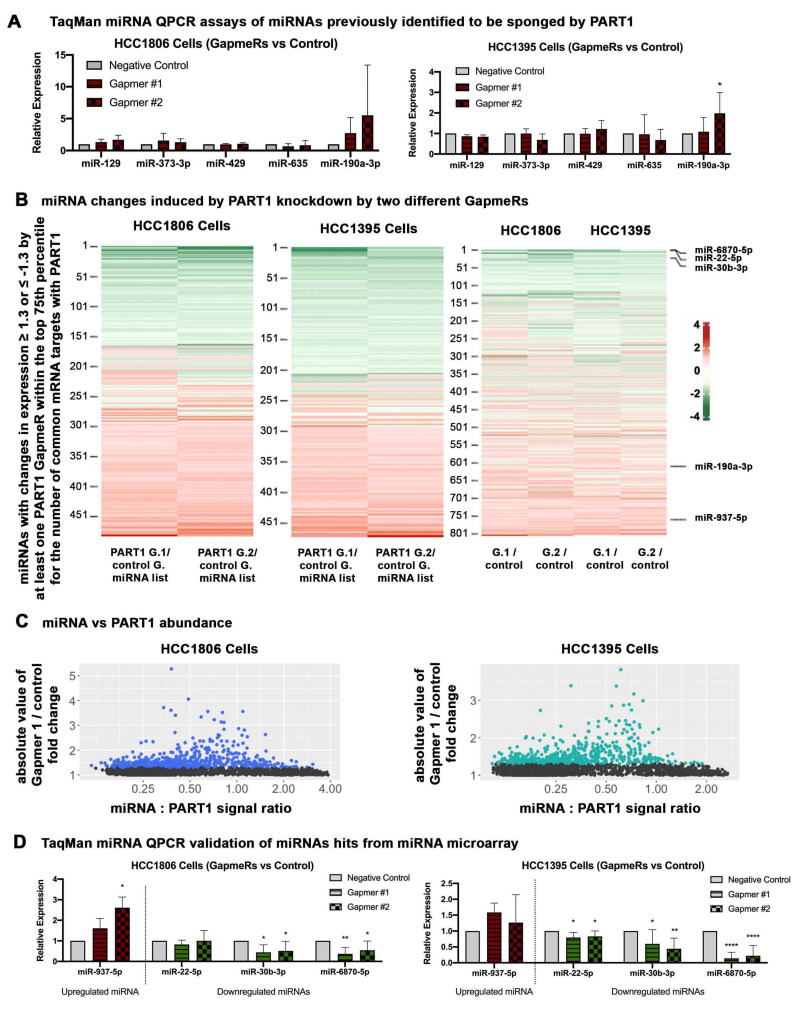
*PART1* knockdown alters the miRNA landscape in HCC1806 and HCC1935 TNBC cells. (**A**) TaqMan miRNA assays of miRNAs previously implicated as being sponged by *PART1* in non-breast cancer cells were assessed in HCC1806 and HCC1935 cells with or without GapmeR-induced knockdown of *PART1* (Gapmer1 (G.1) or Gapmer2 (G.2) versus control GapmeR (control G.) (*n* = 4–8, significance determined by one-way ANOVA, followed by Dunnett’s post-test for multiple comparisons). Error bars represent SD. *miR-129*, *miR-373-3p*, *miR-429* and *miR-635* levels were quantified by the TaqMan miRNA assays and the expression is normalized to reference miRNAs *RNU48* and *miR-221*. *miR-190a-3p* levels were quantified by the TaqMan miRNA Advanced assays and expression is normalized to reference miRNAs *miR-21-5p* and *miR-26b-5p*. (**B**) The heatmaps show miRNAs with an expression fold change ≥1.3 or ≤−1.3 and within the top 75th percentile for the number of common mRNA targets with *PART1* regulated mRNAs (corresponding to at least 15 common genes in HCC1806s cells and 9 in HCC1395 cells) induced by at least one *PART1*-specific GapmeR. (**C**) The abundance of all the miRNAs in the 4.0 miRNA gene chip array relative to *PART1* (abundance extrapolated from Appendix A) detected in the in the negative control samples were calculated for HCC1806 and HCC1395 cells (average of 3n). (**D**) TaqMan miRNA advanced assays of some of the miRNAs identified as being upregulated or downregulated by *PART1* knockdown in HCC1806 of HCC1935 cells in the gene chip array in (**C**) (*n* = 7–8, significance determined by one-way ANOVA, followed by Dunnett’s post-test for multiple comparisons, and expression is normalized to reference miRNAs *miR-21-5p* and *miR-26b-5p*). Significant *p* values are indicated as follows in the figures: *p* < 0.05 = *, *p* < 0.01 = **, *p* < 0.0001 = ****.

**Figure 6 cancers-13-02644-f006:**
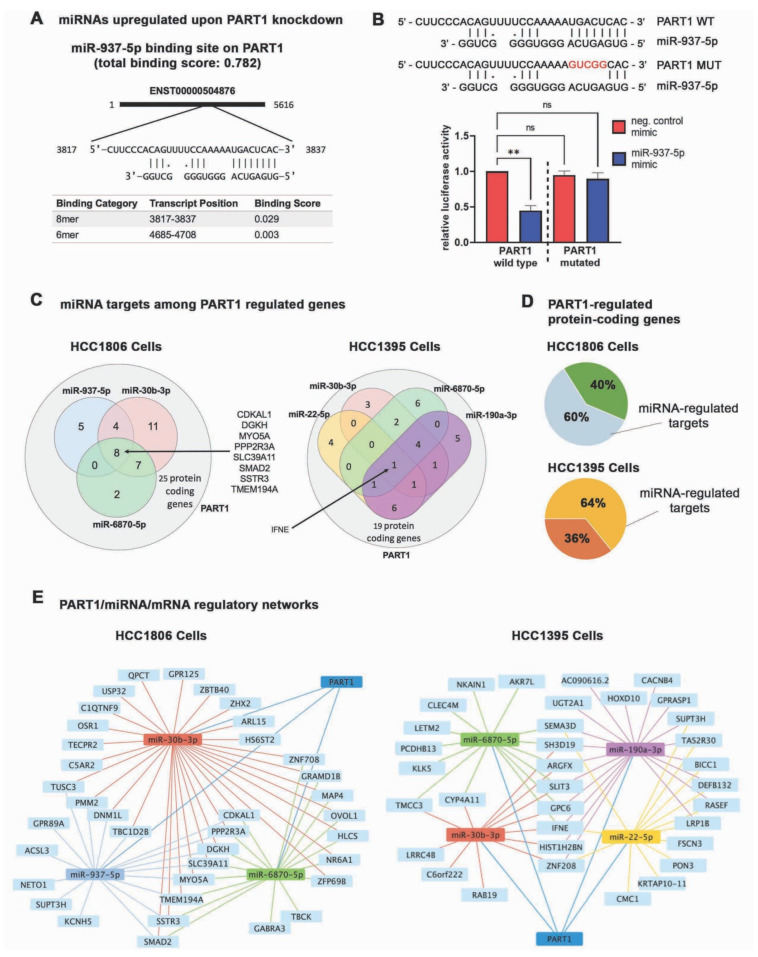
The *PART1*-miRNA-mRNA network in TNBC cells. (**A**) The LncBase v2 predicted *PART1* binding sites and affinity threshold score of miR-937-5p. (**B**) Relative luciferase activity generated by HCC1806 cells transfected with pmirGLO dual-luciferase miRNA target expression vector bearing the predicted *PART1* target sequence for *miR-937-5p* (wildtype, WT) or a mutated version (MUT), and also treated with mimic-hsa-miR-937-5p or mimic negative control (*n* = 3, significance determined by one-way ANOVA, followed by Dunnett’s post-test for multiple comparisons). Luciferase activity is made relative to the cells treated with the mimic negative control and bearing the WT sequence vector. (**C**) Venn diagrams visualize the number of *PART1* regulated genes that are predicted miRNA targets. (**D**) Pie charts depict the proportion of *PART1* regulated mRNAs that are potentially regulated by the miRNAs identified to interact with *PART1*. (**E**) The network node analysis visualizes the *PART1*-regulated miRNAs *miR-937-5p*, *miR-30b-3p* and *miR-6870-5p* in HCC1806 cells and *miR-190a-3p*, *miR-22-5p*, *miR-30b-3p* and *miR-6870-5p* in HCC1395 cells connected with *PART1*-regulated mRNAs. Significant *p* values are indicated as follows in the figures: *p* < 0.01 = **. Non-significant *p* values are either indicated as ns, or not noted.

## Data Availability

The raw data for the Affymetrix Human Gene 2.0 ST and MiRNA 4.0 gene chips have been deposited on GEO (GSE156114 and GSE163569).

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
