# Peer review of "LncRNA PART1 Promotes Proliferation and Migration, Is Associated with Cancer Stem Cells, and Alters the miRNA Landscape in Triple-Negative Breast Cancer"

_cancers, 2021, doi:10.3390/cancers13112644_

Round 1

Reviewer 1 Report

Authors have adequately addressed the queries and it should be accepted with n

Author Response

Thank you!

Reviewer 2 Report

The authors have performed additional experiments and analyses or have provided reasonable explanations for the experiments they didn’t. I have two minor comments:

  1. Please move the methods for the gap closure experiments to the end of section 2.4, where “migration” is included in the heading.
  2. The manuscript, especially the newly added portions, must be proofread for sentence structure and grammar. There are still many such mistakes.

Author Response

Dear Reviewer,

Thank you! We have done as suggested and have proofed the manuscript. We found many typos and identified sentences that needed to be revised to improve clarity. Those corrections are now complete.

This manuscript is a resubmission of an earlier submission. The following is a list of the peer review reports and author responses from that submission.

Round 1

Reviewer 1 Report

In the manuscript, the authors investigate expression and function of lncRNA PART1 in breast cancer. PART1 expression is demonstrated to be high in TNBC and regulated by androgen. PART1 knockdown decreases growth and increases apoptosis in TNBC cells. Correlation of PART1 expression and cancer stemness properties are shown in TNBC cells. In addition, PART1 modulates miRNA profiles in TNBC cells. Each part of experiments is preliminary to support the authors’ conclusion. More consideration described below is required for understanding the expressional regulation, function, and clinical relevance of PART1 in breast cancer.

Major comments:

  1. Concerning Fig. 1, it is not enough to conclude that PART1 is regulated by androgen/AR. AR antagonist and siRNA should be examined in Fig. 1G. More importantly, since interaction between androgen signaling and PART1 is not analyzed in other parts of the manuscript, the role of the PART1 gene as an androgen response gene is unclear in breast cancer.
  2. In Fig. 3, stemness properties are more precisely analyzed using cancer stem cell markers. Do PART1 knockdown and overexpression affect migration, invasion, metastasis, therapeutic drug resistance, and stem cell marker expression?
  3. Concerning Fig. 4c, as there are no whole genomic transcript on the microarray chip, this percentage calculation seems to be incorrect. The percentages should be presented individually against the gene characteristics (protein coding, snRNA, miRNA, and so on).
  4. Concerning Fig. 4e, in situ hybridization is required to confirm the subcellular localization of PART1 transcript. In addition, it is unclear how the values were calculated.
  5. The mechanism of PART1 function is not demonstrated. Can PART1 bind with its target miRNAs?
  6. Almost data were obtained using loss-of-function experiments. To strength the authors’ conclusion, overexpression experiments should be performed in key experiments.

Minor comments:

  1. Statistical analysis comparing control group and treatment groups are required in Figs 2 and 3.
  2. Line 118, “1 x 104” is a typo.

Author Response

We thank the reviewers for their helpful comments for improving the manuscript. We have made changes accordingly (highlighted in the manuscript as red text) and performed additional analyses and experiments to address the suggestions. In the revision, we include multiple new figures (Figs. 1F, 1H, 1J, 2G, 6B), five new supplemental figures (Figs. S1, S2, S3, S4, S5), and revisions to many of the original figures.

REVIEWER 1

Comments and Suggestions for Authors

In the manuscript, the authors investigate expression and function of lncRNA PART1 in breast cancer. PART1 expression is demonstrated to be high in TNBC and regulated by androgens. PART1 knockdown decreases growth and increases apoptosis in TNBC cells. Correlation of PART1 expression and cancer stemness properties are shown in TNBC cells. In addition, PART1 modulates miRNA profiles in TNBC cells. Each part of experiments is preliminary to support the authors’ conclusion. More consideration described below is required for understanding the expressional regulation, function, and clinical relevance of PART1 in breast cancer.

Major comments:

  1. Concerning Fig. 1, it is not enough to conclude that PART1is regulated by androgen/AR. AR antagonist and siRNA should be examined in Fig. 1G. More importantly, since interaction between androgen signaling and PART1 is not analyzed in other parts of the manuscript, the role of the PART1 gene as an androgen response gene is unclear in breast cancer.

Response: We have expanded the analyses as suggested. We treated ER+ T47D cells, which had the greater induction of PART1 upon synthetic androgen treatment (R1881, Fig 1I) with androgen receptor antagonist (D36) alone or in combination with R1881 (new Fig. 1J). Treatment with D36 suppressed the induction of PART1 by R1881 in T47D cells consistent with its regulation by AR.

We assessed the correlation between expression of PART1 and AR and the androgen signaling gene panel (cBioPortal) in breast cancer patient tumors (new Fig. 1F and supplemental Fig. S1). In patient tumors, there is a lack of positive correlation between AR, androgen signaling genes, and PART1 expression. In fact, there is a significant negative correlation between AR and PART1 expression (new Fig. 1F). We find that basal-like (TNBC) has the least expression of AR (Fig. 1G and new Fig. 1H).

Finally, we also assessed expression of AR and the androgen signaling genes in response to PART1 knockdown in TNBC HCC1395 and HCC1806 (new supplemental Fig. S5) and noted minimal effects on androgen signaling genes (please see lines 519-521).  

Together this data suggests that while PART1 is inducible by androgen via the androgen receptor in some breast cancer cells expressing sufficiently high AR (i.e., ER+ breast cancer cells). Overall, this has minimal significance in terms of high PART1 expression in breast cancer since basal-like/TNBC breast cancer patient tumors that have the highest levels of PART1 and the lowest levels of AR.

Please see lines 343-365 where this new data is discussed.

Notably in the breast cancer cell lines, AR is weakly correlated with PART1 and in breast cancer patient tumors AR is significantly negatively correlated with PART1 (Fig. 1E, F). AR expression is lowest in basal-like/TNBC cell lines (e.g., HCC1806 and HCC1395) and highest in luminal/estrogen receptor positive (ER+) cell lines (e.g., T47D, Fig.1G). We similarly noted that basal-like breast cancer patient tumors had the least AR expression (Fig. 1H). This suggests that in the higher AR expressing ER+ breast cancer cell lines cultured in androgen-containing media, PART1 expression may be Fat least partially dependent on androgen signaling. To test this hypothesis, we cultured TNBC HCC1806 cells and ER+ T47D cells in phenol-red free/charcoal stripped FBS, with or without the addition of 10nM synthetic androgen R1881. This resulted in a modest induction of PART1 expression in basal-like HCC1806 cells (1.26-fold, Fig. 1I), and a much more significant induction of PART1 in T47D cells (1.73-fold, Fig. 1I). This is consistent with the higher levels of AR in T47D cells versus HCC1806 cells (Fig. 1G). Addition of AR antagonist D36 inhibited the induction of PART1 by R1881, confirming the role of AR (Fig. 1J). Therefore, PART1 expression can be amplified by the presence of androgens in breast cancer in AR expressing cell lines (e.g., ER+ T47D). To assess the potential clinical relevance of PART1/AR signaling in breast cancer we assessed the correlation of PART1 expression with the 10 gene androgen signaling panel (cBioportal) across breast cancer subtypes in two breast cancer patient tumor datasets (Fig. S1). This failed to reveal strong correlations between androgen signaling genes and PART1 in breast cancer patient tumors. Together this data leads us to conclude that in TNBC/basal-like breast cancer where PART1 expression is highest and most likely clinically relevant, androgen induction does not play a major role.”

  1. In Fig. 3, stemness properties are more precisely analyzed using cancer stem cell markers. Do PART1knockdown and overexpression affect migration, invasion, metastasis, therapeutic drug resistance, and stem cell marker expression?

Response: We have expanded the analyses as suggested. First, we performed gap closure migration assay and noted reduced migration upon PART1 knockdown (new Fig. 2G). We also assessed expression of CSC markers and stemness genes in response to PART1 knockdown in HCC1395 and HCC1806 (new supplemental Fig. S4) and noted minimal effects on CSC markers and stemness genes (please see lines 519-521). Hence it seems that while higher PART1 is associated with breast CSC populations and its knockdown affects the migratory capacity and mammosphere formation capacity, the lncRNA does not directly affect gene regulation associated with CSC populations.

  1. Concerning Fig. 4c, as there are no whole genomic transcript on the microarray chip, this percentage calculation seems to be incorrect. The percentages should be presented individually against the gene characteristics (protein coding, snRNA, miRNA, and so on).

Response: Thank you, we see how this was unclear. We have clarified Fig. 4C by adding a reference bar graph that depicts the percentage of probes in the Affymetrix human gene 2.0ST array which detect the distinct classes of transcripts (see revised Fig. 4C). This more clearly depicts which class of transcripts are altered in abundance upon PART1 knockdown.

  1. Concerning Fig. 4e, in situ hybridization is required to confirm the subcellular localization of PART1 In addition, it is unclear how the values were calculated.

Response: Although it is true that this would provide further information, there are multiple methods that are accepted to map RNA molecules, and each has their advantages and disadvantages in terms of convenience and accuracy. Among the single-gene methods (as is the case for our PART1 analysis) the most commonly used method is QPCR on extracted RNA of purified cell compartment fractions (Wang et al. 2006) as we did in Fig. 4E. This method provides relative transcript concentrations in the compartments but not absolute molecule numbers per cell. In addition to our own analysis to our sub-cellular fractionation analysis of PART1, we also include the validated data from LncAtlas (Mas-Ponte et al. 2017), which provides lncRNA localization across multiple human cell lines based on RNA-sequencing of fractionated cellular compartments. This separate analysis from LncAtlas substantiates our findings in Fig. 4E that PART1 is predominately cytoplasmic (Fig. 4D). The method suggested by the reviewer, fluorescence in situ hybridization (FISH), would provide further proof and can in principle determine absolute numbers of molecules in the cellular compartments (Dunagin et al. 2015); however, it is also a difficult, time-consuming method that requires expensive reagents (Cabili et al. 2015). Given that our method is the most widely used and is accepted, we instead directed our efforts in the revision towards the other suggested experiments. We do acknowledge in the revision that performing FISH would be an additional confirmatory analysis. Please see lines 714-718.

Cabili, M. N., Dunagin, M. C., McClanahan, P. D., Biaesch, A., Padovan-Merhar, O., Regev, A., et al. (2015). Localization and abundance analysis of human lncRNAs at single-cell and single-molecule resolution. Genome Biology, 16(1). https://doi.org/10.1186/s13059-015-0586-4

Clément, T., Salone, V., & Rederstorff, M. (2015). Dual luciferase gene reporter assays to study miRNA Function. Methods in Molecular Biology, 1296, 187–198. https://doi.org/10.1007/978-1-4939-2547-6_17

Dunagin, M., Cabili, M. N., Rinn, J., & Raj, A. (2015). Visualization of lncRNA by single-molecule fluorescence in situ hybridization. Methods in Molecular Biology, 1262, 3–19. https://doi.org/10.1007/978-1-4939-2253-6_1

Jin, Z., Piao, L., Sun, G., Lv, C., Jing, Y., & Jin, R. (2020). Long non-coding RNA PART1 exerts tumor suppressive functions in glioma via sponging mir-190a-3p and inactivation of PTEN/AKT pathway. OncoTargets and Therapy, 13, 1073–1086. https://doi.org/10.2147/OTT.S232848

Mas-Ponte, D., Carlevaro-Fita, J., Palumbo, E., Pulido, T. H., Guigo, R., & Johnson, R. (2017). LncATLAS database for subcellular localization of long noncoding RNAs. RNA, 23(7), 1080–1087. https://doi.org/10.1261/rna.060814.117

Wang, Y., Zhu, W., & Levy, D. E. (2006). Nuclear and cytoplasmic mRNA quantification by SYBR green based real-time RT-PCR. Methods, 39(4), 356–362. https://doi.org/10.1016/j.ymeth.2006.06.010

  1. The mechanism of PART1function is not demonstrated. Can PART1 bind with its target miRNAs?

Response: We have performed additional experiments to address this concern. In Figs. 5 and 6, and Supplemental Fig. S8, we report two potential interactions with PART1 – miR190a-3p and miR-937-5p. The interaction with miR190a-3p has been previously reported and interrogated (Jin et al. 2020), hence we focused our experimental verification on miR-937-5p, which has not been reported previously. In new Fig 6B, we have completed a miRNA luciferase reporter assay (Clément et al. 2015) to assess this novel potential PART1 interaction. This confirmed the binding of PART1 to miR-937-5p. Please see lines 656-665 where this new result is discussed.

 Clément, T., Salone, V., & Rederstorff, M. (2015). Dual luciferase gene reporter assays to study miRNA Function. Methods in Molecular Biology, 1296, 187–198. https://doi.org/10.1007/978-1-4939-2547-6_17

Jin, Z., Piao, L., Sun, G., Lv, C., Jing, Y., & Jin, R. (2020). Long non-coding RNA PART1 exerts tumor suppressive functions in glioma via sponging mir-190a-3p and inactivation of PTEN/AKT pathway. OncoTargets and Therapy, 13, 1073–1086. https://doi.org/10.2147/OTT.S232848

  1. Almost data were obtained using loss-of-function experiments. To strength the authors’ conclusion, overexpression experiments should be performed in key experiments.

Response: We agree that this would be nice data to include; however, we did perform experiments with multiple models, two different GapmeR sequences, and two different shRNA sequences, which all gave similar results. We believe that this does provide evidence of PART1 oncogenic function. In the revision we acknowledge that overexpression experiments would provide further proof and mention this in the Discussion as something that should be done in the future. Please see lines 709-711.

“In the future, experiments where the PART1 transcripts are overexpressed, resulting in increased mammosphere forming potential, proliferation and migration would substantiate these data further.”

Minor comments:

  1. Statistical analysis comparing control group and treatment groups are required in Figs 2 and 3.

Response: Thank you, we agree and we have now added a multiple comparisons post test analysis to each of the assays in Figs. 2 and 3.

  1. Line 118, “1 x 104” is a typo.

Response: We have fixed the typo, thank you! See Line 123.

Reviewer 2 Report

Paola Marcato and colleagues have evaluated the PART1 expression in TNBCs and observed that upregulation of PART1 was associated with worse outcomes. Silencing of PART1 in TNBC cells and PDX model attenuated cellular and tumor growth, mammosphere formation while inducing apoptosis. Transcriptome and miRNA analyses displayed differential expression of miR-190a-3p, miR-937-5p, miR-22-5p, miR-30b-3p, and miR-6870-5p.

This manuscript is written well and data support the scientific conclusion. This manuscript can be published after addressing the following comments

  1. Functionally characterized lncRNAs have been shown to affect gene expression by acting 45 as activators or decoys for transcription factors, recruiters of chromatin-modifying com- 46 plexes, miRNA sponges, and scaffolds of molecular complexes [1]. Author can cite these articles to support this statement. Biochim Biophys Acta Rev Cancer. 2020 Dec;1874(2):188423; Cells. 2020 Jun 21;9(6):1511. 
  2. As far Androgen signaling is concerned. Author should explain whether TNBC cell lines were grown in phenol red. Why they cultured non-TNBC cells in phenol red if they think that androgenic signaling molecules are present in the phenol red and that result in PART1 overexpression.
  3. Author should clarify whether PART1 silencing have any effect on Androgen signaling in their transcriptomics and miRNA analysis.
  4. PART1 knockdown cells into the mammary fat pads of several NOD/SCID mice. Author should provide the exact number of mice used in this study
  5. In figure 2B, PART1 shRNA2 does not seems to have significant effect on proliferation. Please clarify as error bars are touching with control. I suggest to perform cell cycle analysis as well.
  6. In figure 2D the vertical axis showed the volume of tumors and it seems that author can measure very tiny tumors. What was the method. Please clarify
  7. In figure 2G, gapmers seems to be less effective than shRNA what about the effect of gapmers on apoptosis.
  8. I will suggest authors to provide few western blot for apoptosis markers
  9. What is the effect of PART1 gapmers on the mammosphere of non TNBC cells
  10. Please improve the figure 4B panel of qRT-PCR as these are too tiny to undertand

Author Response

We thank the reviewers for their helpful comments for improving the manuscript. We have made changes accordingly (highlighted in the manuscript as red text) and performed additional analyses and experiments to address the suggestions. In the revision, we include multiple new figures (Figs. 1F, 1H, 1J, 2G, 6B), five new supplemental figures (Figs. S1, S2, S3, S4, S5), and revisions to many of the original figures.

REVIEWER 2

Paola Marcato and colleagues have evaluated the PART1 expression in TNBCs and observed that upregulation of PART1 was associated with worse outcomes. Silencing of PART1 in TNBC cells and PDX model attenuated cellular and tumor growth, mammosphere formation while inducing apoptosis. Transcriptome and miRNA analyses displayed differential expression of miR-190a-3p, miR-937-5p, miR-22-5p, miR-30b-3p, and miR-6870-5p.

This manuscript is written well and data support the scientific conclusion. This manuscript can be published after addressing the following comments

  1. Functionally characterized lncRNAs have been shown to affect gene expression by acting 45 as activators or decoys for transcription factors, recruiters of chromatin-modifying com- 46 plexes, miRNA sponges, and scaffolds of molecular complexes [1]. Author can cite these articles to support this statement. Biochim Biophys Acta Rev Cancer. 2020 Dec;1874(2):188423; Cells. 2020 Jun 21;9(6):1511. 

Response: Thank you for the suggestion, we have added the two references.

  1. As far Androgen signaling is concerned. Author should explain whether TNBC cell lines were grown in phenol red. Why they cultured non-TNBC cells in phenol red if they think that androgenic signaling molecules are present in the phenol red and that result in PART1

Response: That is a good point. We have now added our rationale for our choice of media for the functional assays. Please see lines 404-408.

“Since the androgen induction response (although significant) is minimal in TNBC cells (Fig. 1I), and that there is no connection between androgen signaling and PART1 in breast cancer patient tumors (Fig. S1), we opted to not use charcoal stripped FBS and phenol red free media for the functional assays.”

Please also note that as per Reviewer 1’s request, we have also expanded our androgen signalling analysis to include additional breast cancer patient tumor analyses (new Fig. 1F, H, S1), and experiments with androgen receptor antagonist D36 (new Fig. 1J).

  1. Author should clarify whether PART1silencing have any effect on Androgen signaling in their transcriptomics and miRNA analysis.

Response: Good point, we have done this new analysis. We assessed expression of AR and the androgen signaling genes in response to PART1 knockdown in HCC1395 and HCC1806 (new supplemental Fig. S5) and noted minimal effects on androgen signaling genes (please see lines 519-521).

  1. PART1knockdown cells into the mammary fat pads of several NOD/SCID mice. Author should provide the exact number of mice used in this study

Response: We have added the missing information (n=7) to the Materials and Methods section and to the figure legend for Figure 2.

  1. In figure 2B, PART1shRNA2 does not seems to have significant effect on proliferation. Please clarify as error bars are touching with control. I suggest to perform cell cycle analysis as well.

 Response: To confirm the significance of the finding, we have performed additional n (total of 5 in revised figure 2B) and tested for significance by a one-way ANOVA, followed by a Dunnett’s post-test for multiple comparisons). Please see revised Figure 2. We appreciate that cell cycle analysis would be informative but hope that this further analysis, plus the additional new migration gap closure assay we performed (new Figure 2G, requested by Reviewer 1) provides additional functional evidence.

  1. In figure 2D the vertical axis showed the volume of tumors and it seems that author can measure very tiny tumors. What was the method. Please clarify

Response: We used calipers to measure tumor dimensions in the mm range. We grasp the mice, wet the tumor area so that the fur does not obscure the visibility, and measured detectable tumors in the three dimensions, length, width and height. For the PART1 shRNA1 group, two of the 7 did not have detectable tumors, so we scored them as 0 for tumor volume; however, when we performed a necropsy at termination, we detected tumors in all the mice (all 14 mice, 7n per group), hence we show final weights for all tumors in Fig 2C. We originally used the tumor volume calculation of length X width X height /2. The height is the smallest dimension (1mm in some instances, so is more error prone). Given this concern regarding the tumor volume measurement, in the revision we switched to the more commonly used formula of length X width X width /2. This removed the smallest height dimension from the calculation so that the smallest (likely most error prone number is not included). This did not change the result or conclusions.

.

These details have been added to the revision, please see lines 128-135.

Tumor volumes were quantified with caliper measurements (mm3, length x width x width / 2). Mice were grasped, the tumor area was wet so that the fur does not obscure the visibility the tumor, and the tumor was measured in the longest dimension (length, l) and the second longest dimension (width, w). For the PART1 shRNA1 group, two of the 7 did not have detectable tumors, so we scored them as 0 for tumor volume; however, when we performed a necropsy at termination, we detected tumors in all the mice (all 14 mice, 7n per group), hence final tumor weights were determined for all 14 mice.”

  1. In figure 2G,gapmers seems to be less effective than shRNA what about the effect of gapmers on apoptosis.

Response: We have expanded our apoptosis analyses as requested in the revision.  We detected some effects on apoptosis in the PART1 shRNA knockdown clones (only significant in one shRNA when we did multiple comparisons statistical post-test analysis), but none in the GapmeR treated cells and no detectable caspase 3 cleavage by western blotting, leading us to conclude that overall PART1 inhibition has minimal effects on apoptosis (Fig. S2).

Please see lines 444-447, and new Supplemental Figure S2 where these results are described and shown.

  1. I will suggest authors to provide few western blot for apoptosis markers

Response: We performed the western blot on the most well described apoptotic marker, cleavage of caspase 3. This revealed a negative result for all conditions except the included positive control (new Supplemental Figure S2). Together with the additional flow cytometry analyses we performed, we conclude that PART1 knockdown/inhibition has minimal effects on apoptosis (lines 444-448). However, our new gap closure analyses have revealed effects on migration (See new Fig. 2G).

  1. What is the effect of PART1 gapmers on the mammosphere of non TNBC cells

Response: Although this would be interesting to test, since PART1 is lowly expressed in non-TNBC breast cancer cell patient tumors, we did not prioritize this experiment in the revision.

  1. Please improve the figure 4B panel of qRT-PCR as these are too tiny to understand

Response: We have increased the font and legibility of Fig. 4B as requested.

Reviewer 3 Report

Cruickshank et al. use multiple in vivo, in vitro, and omics tools to assess the role of PART1 in TNBC, specifically CSCs and miRNAs. The study is well designed and executed. However, a few key pieces of data are missing.

Major comments:

  1. The authors note that their enumeration of Aldefluorhigh CSC population is important here as CD44/CD24 based populations are also known to be present. In keeping with this, the authors should assess the effect of shRNA/GapmeR-mediated PART1 decrease on the proportion of Aldefluorhigh population in the two cell lines?
  2. To validate their assertion that PART1 inhibition could be a therapeutic strategy for TNBC treatment, the authors must do in vivo experiments wherein they treat established tumors with PART1 and control GapmeRs.

Minor comments:

  1. The manuscript has many spelling and grammar mistakes and several instances of incomplete/awkward sentence constructions. Proofreading and language editing are needed.
  2. Figure 1E, the correlation coefficient should be corrected to, I am assuming, 0.4. Does the correlation persist if only basal-like or claudin-low cells are tested? How do the two cell lines used here compare in terms of AR expression?
  3. Figure 1G, was the androgen treatment experiment carried out in the other cell line, HCC1395?
  4. Are the tumor volumes in Figure 2 correct? 40mm3 would be a very small tumor.
  5. Line 394, the “if” clause is incomplete. Please rewrite.
  6. Figure 4B, did the authors test all genes that were up/downregulated on the microarray gene chip using qPCR? If not, how were these genes picked?
  7. What are the reference genes used in the qPCR analysis? Provide primer details.
  8. Line 507, rewrite the sentence.

Author Response

We thank the reviewers for their helpful comments for improving the manuscript. We have made changes accordingly (highlighted in the manuscript as red text) and performed additional analyses and experiments to address the suggestions. In the revision, we include multiple new figures (Figs. 1F, 1H, 1J, 2G, 6B), five new supplemental figures (Figs. S1, S2, S3, S4, S5), and revisions to many of the original figures.